# miRNA Expression Is Increased in Serum from Patients with Semantic Variant Primary Progressive Aphasia

**DOI:** 10.3390/ijms23158487

**Published:** 2022-07-30

**Authors:** Maria Serpente, Laura Ghezzi, Chiara Fenoglio, Francesca R. Buccellato, Giorgio G. Fumagalli, Emanuela Rotondo, Marina Arcaro, Andrea Arighi, Daniela Galimberti

**Affiliations:** 1Neurodegenerative Diseases Unit, Fondazione IRCCS Ca’ Granda, Ospedale Maggiore Policlinico, 20122 Milan, Italy; francesca.buccellato@unimi.it (F.R.B.); giorgiofumagalli@hotmail.com (G.G.F.); emanuela.rotondo@policlinico.mi.it (E.R.); maria.arcaro@policlinico.mi.it (M.A.); andrea.arighi@policlinico.mi.it (A.A.); daniela.galimberti@unimi.it (D.G.); 2Department of Neurology, Washington University School of Medicine, St. Louis, MO 63110, USA; lghezzi@wustl.edu; 3Department of Pathophysiology and Transplantation, Dino Ferrari Center, University of Milan, 20122 Milan, Italy; chiara.fenoglio@unimi.it; 4Department of Biomedical, Surgical and Dental Sciences, Dino Ferrari Center, University of Milan, 20122 Milan, Italy

**Keywords:** circulating miRNAs, semantic variant primary progressive aphasia (svPPA), logopenic variant PPA, gene expression

## Abstract

Primary progressive aphasia (PPA) damages the parts of the brain that control speech and language. There are three clinical PPA variants: nonfluent/agrammatic (nfvPPA), logopenic (lvPPA) and semantic (svPPA). The pathophysiology underlying PPA variants is not fully understood, including the role of micro (mi)RNAs which were previously shown to play a role in several neurodegenerative diseases. Using a two-step analysis (array and validation through real-time PCR), we investigated the miRNA expression pattern in serum from 54 PPA patients and 18 controls. In the svPPA cohort, we observed a generalized upregulation of miRNAs with miR-106b-5p and miR-133a-3p reaching statistical significance (miR-106b-5p: 2.69 ± 0.89 mean ± SD vs. 1.18 ± 0.28, *p* < 0.0001; miR-133a-3p: 2.09 ± 0.10 vs. 0.74 ± 0.11 mean ± SD, *p* = 0.0002). Conversely, in lvPPA, the majority of miRNAs were downregulated. GO enrichment and KEGG pathway analyses revealed that target genes of both miRNAs are involved in pathways potentially relevant for the pathogenesis of neurodegenerative diseases. This is the first study that investigates the expression profile of circulating miRNAs in PPA variant patients. We identified a specific miRNA expression profile in svPPA that could differentiate this pathological condition from other PPA variants. Nevertheless, these preliminary results need to be confirmed in a larger independent cohort.

## 1. Introduction

The term primary progressive aphasia (PPA) refers to a diverse group of dementias that present prominent and early problems with speech and language, often occurring before the age of sixty-five. The current diagnostic criteria [1] recognize three clinical-anatomical syndromes of PPA: nonfluent/agrammatic variant PPA (nfvPPA), logopenic variant PPA (lvPPA) and semantic variant PPA (svPPA), each of which is characterized by specific linguistic deficits and different neuroanatomical involvement. The syndromes nfvPPA and svPPA are classified as frontotemporal dementia (FTD) syndromes because of their propensity to affect the frontal and temporal lobes, respectively, and their association with frontotemporal lobar degeneration (FTLD) pathology, whereas lvPPA is most commonly viewed as an atypical variant of Alzheimer’s disease (AD) [1,2]. The major histopathological associations differ between the major PPA syndromes: nfvPPA is most associated with FTLD tauopathies, svPPA is closely associated with TDP-43 pathology and lvPPA with amyloid deposition, even if there is not complete concordance between phenotypes and pathology [3]. Based on histopathology, the accumulation of TDP-43 in the brain cortex is classified into five types, type A through E, depending on the forms of TDP-43 accumulation in the cerebral cortex. Type A pathology is frequently observed in FTD and nfvPPA. Type B pathology occurs in FTLD with Amyotrophic Lateral Sclerosis (ALS) and ALS with lesions that extend to the cerebral cortex. Type C pathology is mainly associated with svPPA [4,5].

A lack of understanding of the underlying pathophysiology, along with the difficulties of clinical differentiation of PPA variants, makes the diagnostic procedures rather challenging [3,6]. Excluding off-label cholinomimetics for lvPPA (with positive AD biomarkers), no disease-modifying treatments for PPA variants exist [7]. In the framework of precision medicine, the lack of disease modification approaches emphasized the need to discover useful molecular biomarkers, not only to improve the prediction of molecular pathology, but also to monitor the variant specific response to treatment. In this scenario, micro (mi)RNAs have been considered as key regulators of pathogenesis of several neurodegenerative diseases [8]. miRNAs are single-stranded, non-coding small RNAs that are abundant in plants and animals and are conserved across species. The raw transcripts undergo several nuclear and cytoplasmic post-translation processing steps to generate mature, functional miRNAs. In the cytoplasm, mature miRNAs associate with other proteins to form the RNA-induced silencing complex (RISC), enabling the miRNA to imperfectly pair with cognate miRNA transcripts. The target mRNA is then degraded by the RISC, preventing its translation into protein. miRNA-mediated repression of translation is involved in many cellular processes, such as differentiation, proliferation and apoptosis, as well as other key cellular mechanisms [9]. It is now well established that altered RNA processing could act as a contributing factor to several neurological conditions including aging-related neurodegenerative diseases such as AD, FTD, ALS and Parkinson’s disease (PD) [10]. For instance, in AD, the implication of miRNAs in Aβ production, via BACE1 modulation, and in tau phosphorylation that leads to hyperphosphorylated neurofibrillary tangle formation, has been demonstrated [9]. Altered miRNA signatures were also identified in AD and FTD. In particular, several miRNAs have been identified and differentially expressed in postmortem tissue, blood, and cerebrospinal fluid (CSF) that differ by disease stage [11]. Since miRNAs are able to circulate freely in blood and other body fluids, they are proposed as possible useful biomarkers for several neurodegenerative diseases such as AD [12], PD [8] and multiple sclerosis (MS) [13]. Currently, there are few studies regarding miRNAs’ role in FTD, particularly regarding PPA variants as pathogenic modulators as well as biomarkers [14,15,16].

Herein, we performed an exploratory study of circulating miRNAs in serum from well-characterized PPA variant patients in order to discover possible miRNA signatures and be able to differentiate among PPA variants and/or identify possible new peripheral biomarkers.

## 2. Results

The expression profile of 84 miRNAs was performed in a discovery cohort consisting of 6 svPPA patients, 6 lvPPA, 6 nfv PPA and 6 healthy controls. The volcano plot showed a generalized upregulation of serum miRNAs in svPPA patients compared with healthy subjects (Figure 1A). The statistical analysis led to the identification of two significant upregulated miRNAs: miR-106b-5p (+7.5-fold regulation over controls, *p* = 0.0027) and hsa-miR-133a-3p (+22.5-fold regulation over controls, *p* = 0.000241). Upregulation of these two miRNAs were subsequently confirmed by Taqman qRT-PCR assays in a validation cohort composed of 12 svPPA, 12 lvPPA, 12 nfvPPA patients and 12 healthy controls, confirming previous data (miR-106b-5p: 2.69 ± 0.89 mean ± SD vs. 1.18 ± 0.28, *p* < 0.0001; miR-133a-3p: 2.09 ± 0.10 vs. 0.74 ± 0.11 mean ± SD, *p* = 0.0002, Figure 1B). No differences were observed stratifying data according to gender and age at onset.

A receiver operating characteristic curve (ROC) analysis was performed to evaluate the diagnostic potential of miR-133a-3p and miR-106b-5p. In serum, miR-133a-3p together with miR-106b-5p had an AUC of 0,84 to distinguish svPPA from controls (95% CI: 0.7309 to 0.9636, *p* < 0.001, Figure 2A). Moreover, these two miRNAs had an AUC of 0.9696 to distinguish svPPA from other PPA variants (lvPPA + nfvPPA; 95% CI: 0.9338 to 1000, *p* < 0.001, Figure 2B). Also considering the presence of positive CSF AD biomarkers, serum miR-133a-3p and miR-106b-5p had an AUC of 0,83 to distinguish AD-negative svPPA patients from AD-positive with other PPA variants (AD-positive lvPPA + AD-positive nfvPPA; CI: 0.6576 to 1,000, *p* = 0.0056, Figure 2C).

In nfvPPA and lvPPA patients, deregulation of serum miRNAs expression was observed (Figure 3A,B), although the statistical threshold was not reached. In particular, in lvPPA patients miR-122-5p, miR-21-5p, miR-223-3p, miR-25-3p, miR-22-3p and miR-24-3p showed a strong downregulation (−41-, −18-, −6.54-, −22- and −5.71-fold regulation over controls, *p* > 0.05), whereas miR-15a-5p, miR-124-3p were upregulated (+6.39- and +3.17-fold regulation over controls, respectively, *p* > 0.05).

In order to gain an understanding of the potential biological significance of these miRNA deregulations, an in silico approach to identify potential target genes of miR-133a-3p and miR-106b-5p was used. Two publicly available bioinformatic tools were used: TargetScan 7.0 and miRDB. As shown in Figure 4A, a total number of 492 overlapping predicted target genes for these miRNAs were identified from these algorithms (Appendix A). To understand the functional roles and mechanisms of identified target genes, GO and KEGG analyses were performed using DAVID (Frederick, MD, USA). The results demonstrated that genes were enriched in ten BP, such as, nervous system development and positive regulation of dendrite extension (Figure 4C), while for MF, the genes were enriched in small GTPase binding and in protein binding (Figure 4D). Additionally, CC analysis showed that the genes were enriched, among others, in nucleoplasm, cytoplasm and glutamatergic synapses (Figure 4B). The KEGG pathway analysis predicted that these mRNAs were mainly involved in a MAPK signaling pathway, dopaminergic synapsis and longevity regulating pathway (Figure 4E).

## 3. Discussion

In this study, we showed, for the first time, that circulating miR-106b-5p and miR-133a-3p are upregulated in svPPA but not in other PPA variants, although in a quite small but well characterized cohort of patients. Of note, ROC curves indicated that serum miR-133a-3p and miR-106b-5p have an adequate sensitivity and specificity to distinguish svPPA from controls as well as other PPA variants, and they could be considered good biomarker candidates to aid PPA diagnosis.

Notably, we described a striking different deregulation of various miRNAs in the serum of patients with lvPPA and svPPA, with a generalized downregulation in lvPPA versus an upregulation of circulating miRNAs expressing svPPA. lvPPA has evidence of amyloid deposition in 86% of cases [17], and also CSF amyloid levels in our patients (Table 1) are in line with these data. Therefore, despite being classified clinically as PPA, this entity may share pathogenic mechanisms with AD. Of note, our data show a deregulation, not only of miR-223-3p that is widely researched in AD [18], but also of miR-125b-5p, miR-24-3p and miR-21-5p. These molecules that are downregulated in lvPPA patients (although not reaching statistical significance) are already associated with AD suggesting their possible role in the amyloid metabolism defect [19,20,21,22].

svPPA has been consistently associated with TDP-43 type C pathology [23]. Albeit the dysfunction of TDP-43 has been associated with multiple neurodegenerative diseases, and the exact pathophysiological mechanisms still need to be elucidated. A common pathologic hallmark encompassing all the TDP-43 “proteinopathies” is the cytoplasmic aggregation and nuclear clearance of TDP-43 [24]. This commonality led to the suggestion that nuclear loss-of-function and cytoplasmic gain-of-function could be at the origin of TDP-43 driven neurodegeneration [24]. The lack of nuclear TDP-43 determines the deregulation of RNA synthesis, autophagy and stress granule formation, whereas the pathologic accumulation of insoluble TDP-43 in the cytoplasm leads to deposit formation and ultimately, neuronal death. Mitochondrial dysfunction has been proposed as an additional route mediating TDP-43 pathology [25]. Of note, increased miR-106b-5p, which was significantly upregulated in svPPA, has been associated with the disruption of mitochondrial fusion and the impairment of oxidative phosphorylation [26]. The over-expression of miRA-106b-5p levels in people with svPPA could reflect a tentative compensatory mechanism or just the consequence of mitochondrial damage. Defective microglia clearance of TDP-43 has been proposed as another possible mechanism underlying TDP-43 accumulation [27]. Interestingly, miR-133a-3p, which was significantly upregulated in our cohort of svPPA, has been demonstrated to suppress microglia activation in a mouse model of neuropathic pain [27], potentially favoring TDP-43 accumulation and neurodegeneration. Importantly, several studies underlie the role of miR-133a-3p in the modulation of inflammation. This miRNA is involved, for example, in a competing endogenous RNA (ceRNAs) network. The CircHelz RNA, a circular RNA encoded by helicase with the zinc finger (Helz) gene, promotes the activation of NLPR3 inflammasome by the inhibition of miR-133a-3p expression [28].

miR-106b-5p and miR-133a-3p have recently been associated with other neurodegenerative diseases and proposed as possible biomarkers underling the fact that they are possibly important players in cerebral physiopathology. For instance, deregulated miR-106-5p provided the best discrimination between PD and progressive supranuclear palsy (PSP) patients [29]. Moreover, a downregulation of circulating miR-106b-5p in MS was found, and this data is correlated with increased production of neuroprotective brain derived neurotrophic factor (BDNF) as a compensatory mechanism [30]. Regarding miR-133a-3p, a possible link between upregulated circulating miRNA levels and ALS patients was found, even if ALS is associated with TDP-43 type A and/or B, but not C, which is the TDP-43 subtype often associated with svPPA [31,32]. It is important to take into account that a subset of svPPA cases can present with other pathologies, including AD pathology, which has been described in up to 25% of svPPA [33,34]. Nevertheless, a recent meta-analysis showed that amyloid has been detected (via CSF biomarkers, Amyloid PET or autopsy) in 16% of svPPA [17], which is in line with our results (11%).

Our in silico analysis of miRNA pathways showed that 492 mRNA targets are common between miR-133a-3p and miR-106b-5p, and some of which are involved in neurodegenerative pathways. In this regard, it is worth noting that one of the 492 overlapped mRNAs targets is the solute carrier family 41 member 1 (*SLC41A1*). A recent work reported a rare loss-of-function variant of this gene in a cohort of 80 patients diagnosed with early onset of PD, and one of the mechanisms hypothesized behind the loss of function is that SLC41A1 could influence mitochondrial processes involved in energy production [35,36]. Further, an extensive RNA sequencing study in the frontal cortex tissue of patients with frontotemporal lobar degeneration revealed a decreased differential expression of the small integral membrane protein 14 gene (*SMIM14*) [37]. *SMIM14*, one of the overlapped mRNAs’ target and between our significantly differential regulated miRNAs, is also associated with mitochondrial functions [37]. Taking into consideration that miRNAs are implicated in gene post-transcriptional control and the upregulation of miRNAs can result in the downregulation of target gene expression, and we can speculate that the upregulation of miR-133a-3p and miR-106b-5p may downregulate some of the common 39 target mRNAs. Interestingly, among the 39 common mRNAs targets for miR-133a-3p and miR-106b-5p, there is a *DERL2* gene, encoding for Derlin-2 protein, one of the endoplasmic reticulum (ER) membrane proteins and part of the ERAD degradation complex which eliminates unfolded proteins [38]. Recent findings by Sugiyama et al. demonstrated that Derlin-2-deficient mice have widespread postnatal brain atrophy in the cerebellum and striatum as well as reduced neurite outgrowth and motor function deficits [39]. We can hypothesize that a generalized downregulation of miRNAs observed in lvPPA could lead to a translation upregulation and an alteration of target gene expression [40] and vice-versa in svPPA. However, further studies are needed to clarify how miRNAs target mRNAs in the molecular pathology behind svPPA and lvPPA.

Certainly, a limitation of this study is the small size of the cohort analyzed, which can influence the significance threshold. Therefore, further analyses of miRNA strongly up- or downregulated would be needed, and possibly in comparison to AD. Nevertheless, data obtained in svPPA in such a small cohort suggest that the upregulation of miRNA is actually remarkable in these patients, and the mechanisms at the basis of these observations need to be better studied.

In conclusion, we showed that there is a specific miRNA expression profile in svPPA that differentiates this pathological condition from other PPA variants, and in particular from lvPPA. miR-133a-3p and miR-106b-3p are significantly upregulated in svPPA and also in the validation cohort, but these preliminary results need to be confirmed in a larger independent cohort.

## 4. Materials and Methods

### 4.1. Population

Seventy-two subjects (18 svPPA, 18 lvPPA, 18 nfvPPA and 18 healthy controls) were recruited at the Alzheimer Unit of the Fondazione Cà Granda, IRCCS Ospedale Maggiore Policlinico, University of Milan (Milan). All patients underwent a standard battery of examinations, including their medical history, a physical and neurological examination, laboratory screening tests, neurocognitive evaluation and imaging. The Clinical Dementia Rating (CDR), the Mini Mental State Examination (MMSE), the Frontal Assessment Battery (FAB), the Wisconsin Card Sorting Test (WCST), and the Tower of London test assessed cognitive dysfunctions. The presence of significant vascular brain damage was excluded (Hachinski Ischemic Score < 4). Lumbar punctures were performed after one night of fasting. The PPA patients were diagnosed according to current consensus criteria [1]. All PPA patients were screened for possible mutations in *MAPT*, *GRN* and *C9ORF72* genes. The control group consisted of 18 non-demented volunteers matched for ethnic background and age and without memory and psycho-behavioral dysfunctions (MMSE ≥ 28). Informed consent to participate in this study was given by all subjects or their caregivers. The study protocol had been previously approved by the local Institutional Review Board.

The characteristics of patients and controls are summarized in Table 1.

### 4.2. Serum Sample Collection

Whole blood samples were allowed to sit at room temperature for a minimum of 30 min and a max of 2 h after collection. Separation of the clot was done by centrifugation at 10 min 1300× *g* at room temperature for 15–20 min. The serum samples were removed and dispensed in aliquots of 400 µL and put into cryotubes. Specimens were stored at −80 °C until used.

### 4.3. AD CSF Biomarkers Measurement

CSF samples were collected into 15 mL polypropylene tubes by LP in the L3/L4 or L4/L5 interspace. The samples were centrifuged at 2000 r/min for 10 min at 4 °C. The supernatants were stored at −80 °C until used. CSF Aβ, tau and Ptau were measured with ELISA kits (Fujirebio, Ghent, Belgium). The normal values of biomarkers were: Aβ > 600 pg/mL; tau < 400 pg/mL and Ptau < 61 pg/mL [41].

### 4.4. miRNA Isolation from Human Serum Samples

200 µL of human serum was thawed on ice and lysed with 1.5 volume of Qiazol (Qiagen, Hilden, Germany). To allow sample-to-sample normalization, synthetic C. elegans miRNA cel-miR-39 (Qiagen) was added (as a mixture of 25 fmol of each oligonucleotide in 5 µL total volume) to each denatured sample. RNA was isolated using miReasy Mini Kit, following the manufacturer’s protocol for liquid samples (Qiagen) [12].

### 4.5. Screening of miRNAs in Human Serum by Human Serum & Plasma miScript miRNA PCR Array

RNA was retro-transcribed with miScript II RT kit (Qiagen), according to the instructions of the manufacturer. For real-time PCR experiments, the Human Serum & Plasma miScript miRNA PCR Array (MIHS-106ZC) was used, and runs were performed in a QuantStudio 12K system. The array profiles the expression of 84 of the most abundantly expressed and best characterized miRNA sequences in serum, plasma and other body fluids. These arrays included short non-coding RNAs (SNORD61, SNORD68, SNORD72, SNORD95, SNORD96A and RNU6B-2) for the proper normalization of the data, miRNA reverse transcription control and a positive PCR control.

### 4.6. Validation of Best Hits by Taqman qRT-PCR Assays

3 µL of RNA was reverse transcribed using the TaqMan miRNA Reverse Transcription Kit and miRNA-specific stem-loop primers (ThermoFisher Scientific, Waltham, MA, USA) in a small-scale RT reaction. 0.8 µL of RT product was used for qRT-PCR and was run on QuantStudio 12K using specific TaqMan miRNA assays (ThemoFisher Scientific) [12].

### 4.7. Target Gene Prediction and Bioinformatics Analysis

Target genes of two candidate miRNAs were predicted using online bioinformatic tools; TargetScan 7.0 (available online: http://www.targetscan.org, accessed on 14 July 2022) and miRDB (available online: http:/www.mirdb.org, accessed on 14 July 2022). Venn diagrams (available online: http:/bioinformatics.psb.urgent.be/webtools/Venn, accessed on 14 July 2022) were used to obtain overlapping target genes from the two bioinformatic tools. GO enrichment analyses of biological processes (BP), cellular component (CC), molecular function (MF), Kyoto Encyclopedia of Genes and Genomes (KEGG) and interactome pathways were obtained using the DAVID (available online: http://david.ncifcrf.gov, accessed on 14 July 2022) bioinformatics tool [42]. A false discovery rate (FDR) < 0.05 was used as the cut-off criteria.

### 4.8. Statistical Analysis

The Qiagen PCR Array data analysis was based on ΔΔCt method with normalization of the raw data to housekeeping genes (using the software available at http://www.sabiosciences.com/pcarraydataanalysis.php, accessed on 1 June 2022). *p* values were calculated based on a student’s *t*-test of the replicate 2^(−Delta Ct) values for each gene in the control and PPA groups. Best hits were chosen based on statistical significance (*p* < 0.05). Data distribution was tested for normality with the Kolmogorov–Smirnov and Shapiro–Wilk Test. All data are shown as mean ± SD. The differences between the three groups were analyzed with One-Way ANOVA followed by Tukey’s test as multiple comparison post hoc. Receiver-operating-characteristic (ROC) curves were generated and areas under the curves (AUCs) were calculated as well as specificities for fixed sensitivity values with corresponding 95% confidence intervals (CI). For statistical analysis, GraphPad Prism 9.0 (La Jolla, CA, USA) software was used.

## Figures and Tables

**Figure 1 ijms-23-08487-f001:**
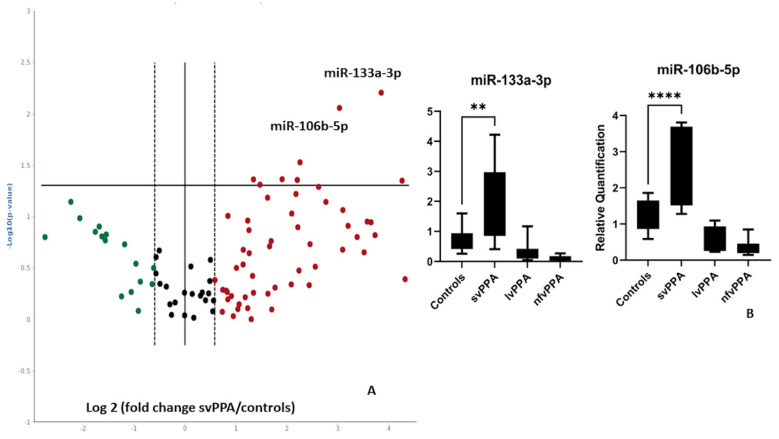
(**A**) Volcano plot of semantic variant PPA versus controls. Data are expressed as Log 2 (fold change) and each dot represents one miRNA. Green indicates downregulation, red upregulation and black dots represent either miRNA that are below the fold change cutoff (**B**) Box plots of miR-103a-3p and miR-106b-5p serum expression levels in svPPA patients compared to controls in the validation cohort. **** *p* < 0.0001, ** *p* = 0.0002.

**Figure 2 ijms-23-08487-f002:**
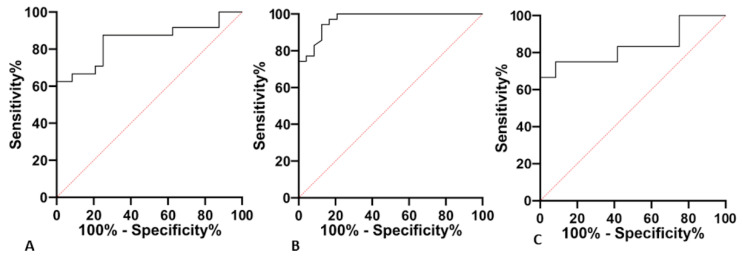
The receiver operating characteristic curve (ROC) results of serum miR-133a-3p together with miR-106b-5p. (**A**) AUC = 0.84, *p* < 0.001 svPPA vs. controls; (**B**) AUC = 0,95, *p* < 0.001 svPPA vs. lvPPA + nfvPPA0; (**C**) AUC = 0,83, *p* = 0.0056 AD-negative svPPA vs. AD-positive lvPPA + AD-positive nfvPPA.

**Figure 3 ijms-23-08487-f003:**
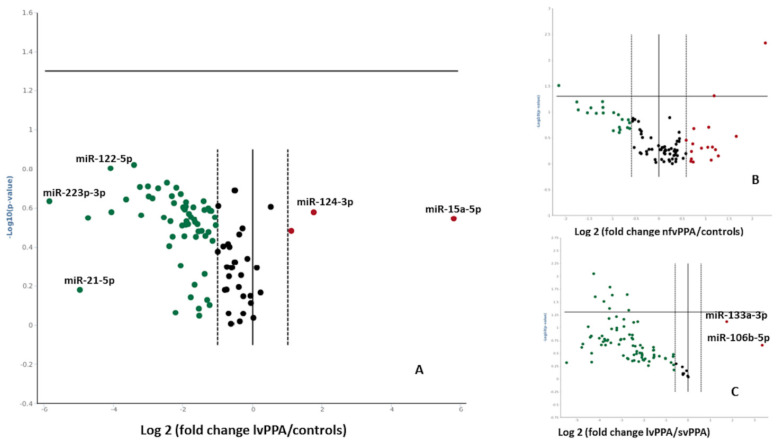
Volcano plots (**A**) lvPPA versus controls (**B**) nfvPPA versus controls (**C**) lvPPA versus svPPA. Data are expressed as Log 2 (fold change) and each dot represents one miRNA. Green indicates downregulation, red upregulation and black dots represent either miRNA that are below the fold change cutoff.

**Figure 4 ijms-23-08487-f004:**
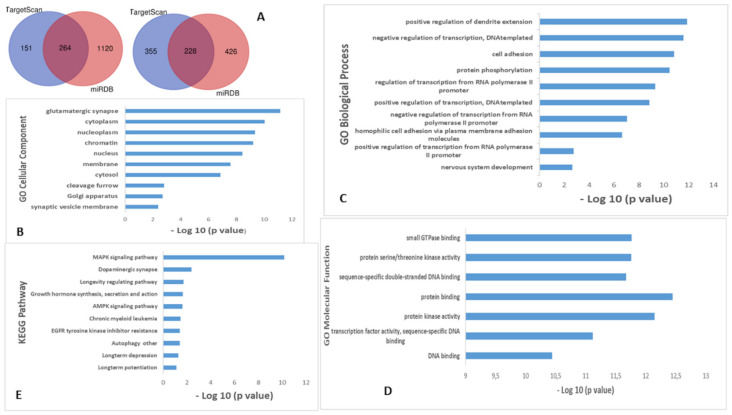
miRNA target and pathway prediction for miR-133a-3p and miR-106b-5p. (**A**) Venn diagram of target genes of two significant svPPA miRNAs using two bioinformatic tools; (**B**) GO enrichment analysis demonstrated the top 10 genes enriched in CC, (**C**) BP, (**D**) MF and (**E**) KEGG pathway analysis.

**Table 1 ijms-23-08487-t001:** Characteristics of PPA patients and controls.

	Nonfluent/Agrammatic PPA (*n* = 18)	Logopenic PPA (*n* = 18)	Semantic Variant PPA (*n* = 18)	Healthy Controls (*n* = 18)
gender (M:F)	6:12	8:10	9:9	10:8
Aβ42 (pg/mL; mean ± SD)	723.6 ± 170.59 *	571 ± 433.7 **	893.15 ± 325.6 ***	1054.54 ± 252
tau (pg/mL; mean ± SD)	491.4 ± 177.50	456.4 ± 177.9	338.15 ± 170.2	380 ± 274
p-tau (pg/mL; mean ± SD)	55.6 ± 25.88	61.4 ± 24.27	49.38 ± 19.05	58 ± 46
mean age (years ± SD)	76.3 ± 8.6	81 ± 8.1	79 ± 9.4	78 ± 7.3
mean age at onset (years ± SD)	63.5 ± 0.44	70 ± 0.36	64 ± 0.98	

Presence of AD signature in * 2, ** 8 and *** 2 patients.

## Data Availability

The data that support the findings of this study are available from the corresponding author upon reasonable request.

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
