# Peer review of "miRNA Expression Is Increased in Serum from Patients with Semantic Variant Primary Progressive Aphasia"

_ijms, 2022, doi:10.3390/ijms23158487_

Round 1

Reviewer 1 Report

Review – MiRNA expression is increased in serum from patients with se-mantic variant Primary Progressive Aphasia

In this article, Serpente et al. examined the miRNA expression pattern in serum from a small sample of PPA patients. They found upregulated miRNAs in svPPA patients, and a downregulation of miRNAs in lvPPA. Subsequent analyses showed that miR-106b-5p and miR-133a-3p miRNAs are involved in different biological pathways and could be relevant for the pathogenesis of neurodegenerative diseases.

First, I would like to congratulate the authors on their work. The research question they aimed to answer with this project is original. However, I have concerns about the general importance of the findings in the context of the diagnosis of primary progressive aphasia. Among others, I believe rewriting the discussing and bonifying the figures would be helpful. Please find below several minor and major comments/questions for the authors.

Introduction:

·      This section is clear and concise. My only comment would be to expand the paragraph on MiRNAs. As someone who is not familiar with these molecules, I am still confused, after reading the introduction, about 1. the role(s) of MiRNAs in neurodegenerative diseases, 2. what previous findings have shown and 3. what this new study adds to the literature. Because the focus is on PPAs, I think this paper will be interesting to a diverse audience, including clinicians, who might not be very familiar with this concept.

Materials and methods:

·      Very clear.

Results:

·      Figure 1 is very hard to read and could be bonified. Could the authors please make it bigger? Moreover, the data could be much better presented (A,B,C under each part of the figure is in very small font; section B is in a rectangle with a title with no capital letters, x axis very hard to read, y axis is not specified, there are unnecessary lines behind the graph, it is written “semantic dementia” in Figure 1C instead of svPPA, the bars are not the same colors and not consistent between the two figures, etc.).

·      Figures 2 and 3: same comment. The figures are not in a presentable format at the moment.

Discussion:

·      The organization of the discussion is not clear to me and should be proofread. After reading this section, I cannot say I clearly understand the novelty of the study and importance of the findings. It is also hard to understand why the authors chose to discuss certain findings (ex: how is the paragraph on miR-106-5p in PD relevant to the current study?). In general, I think the discussion needs a thorough rewriting. A few questions to help the authors rewrite this section: what is the main message and main findings of this paper? How do their results integrate with previous findings? How will their results be helpful in clinics and in the diagnosis of PPAs, in clinical trials and in understanding the pathophysiology (and the heterogeneity) of the different variants of PPAs?

Author Response

Point by point replies to criticisms raised by Reviewer #1

Introduction:

  • This section is clear and concise. My only comment would be to expand the paragraph on MiRNAs. As someone who is not familiar with these molecules, I am still confused, after reading the introduction, about 1. the role(s) of MiRNAs in neurodegenerative diseases, 2. what previous findings have shown and 3. what this new study adds to the literature. Because the focus is on PPAs, I think this paper will be interesting to a diverse audience, including clinicians, who might not be very familiar with this concept.

      We added a paragraph about the role of miRNAs in neurodegenerative diseases in an attempt to clarify the three issues raised: “miRNAs are single stranded, non-coding small RNAs that are abundant in plants and animals, and are conserved across species. The raw transcripts undergo several nuclear and cytoplasmic post-translation processing steps to generate mature, functional miRNAs. In the cytoplasm, mature miRNAs associate with other proteins to form the RNA-Induced Silencing Complex (RISC), enabling the miRNA to imperfectly pair with cognate miRNA transcripts. The target mRNA is then degraded by the RISC, preventing its translation into protein. miRNA-mediated repression of translation is involved in many cellular processes, such as differentiation, proliferation, and apoptosis, as well as other key cellular mechanisms [9]. It is now well established that altered RNA processing could act as a contributing factor to several neurological conditions including aging-related neuro-degenerative diseases such as AD, FTD, ALS and Parkinson’s disease (PD) [10]. For in-stance, in AD, the implication of miRNAs in Aβ production, via BACE1 modulation, and in tau phosphorylation, that leads to hyperphosphorylated neurofibrillary tangle formation, has been demonstrated [9]. Altered miRNA signatures were also identified in AD and FTD. In particular, several miRNAs have identified differentially expressed in post-mortem tissue, blood, and Cerebrospinal Fluid (CSF) that differ by disease stage [11]”.

- Results:

  • Figure 1 is very hard to read and could be bonified. Could the authors please make it bigger? Moreover, the data could be much better presented (A,B,C under each part of the figure is in very small font; section B is in a rectangle with a title with no capital letters, x axis very hard to read, y axis is not specified, there are unnecessary lines behind the graph, it is written “semantic dementia” in Figure 1C instead of svPPA, the bars are not the same colors and not consistent between the two figures, etc.). Figures have been modified according to all comments from Reviewer 1 (and 2).  The Figure numbering has been changed.  

  • Figures 2 and 3: same comment. The figures are not in a presentable format at the moment. We have adjusted Figures as suggested by the Reviewer

- Discussion:

·      The organization of the discussion is not clear to me and should be proofread. After reading this section, I cannot say I clearly understand the novelty of the study and importance of the findings. It is also hard to understand why the authors chose to discuss certain findings (ex: how is the paragraph on miR-106-5p in PD relevant to the current study?). In general, I think the discussion needs a thorough rewriting. A few questions to help the authors rewrite this section: what is the main message and main findings of this paper? How do their results integrate with previous findings? How will their results be helpful in clinics and in the diagnosis of PPAs, in clinical trials and in understanding the pathophysiology (and the heterogeneity) of the different variants of PPAs? Thank you for this comment. We re-wrote some parts of the discussion, taking into account the Reviewer suggestions. First, we tried to underline the main findings of the study: “In this study, we showed, for the first time, that circulating miR-106b-5p and miR-133a-3p are up-regulated in svPPA but not in other PPA variants although in a quite small but well characterized

Reviewer 2 Report

Serpente et al. present data on serum micro RNA (miRNA) expression profiles in a clinically well-characterized cohort of primary progressive aphasia (PPA) patients (18 nonfluent variant (nfvPPA), 18 logopenic variant (lvPPA) and 18 semantic variant (svPPA)) and 18 healthy controls. The rationale for this study was that the underlying pathophysiology in PPA is incompletely understood, including the role of miRNAs. Serum samples were subdivided into a discovery versus a validation cohort (per clinical group: n=6, n=12, respectively). The validation cohort was able to confirm expression patterns of the discovery cohort, using micro-array and real-time PCR. Generally, an upregulation of miRNA was found in svPPA, while in lvPPA, downregulation of miRNA was observed. Pathway analyses was restricted to the svPPA group and indicated that one of the top hits: miR-133a-3p was involved in ECM-receptor-interaction, EGFR and adrenergic pathways, while the other major hit, miR-106b-5p, appeared associated with prion disease, TGF-beta, FOxO signalling and cell cycle processes.

Major points

-In this cohort of PPA patients, the methods and Table 1 describe that patients underwent lumbar puncture for AD biomarker analyses (i.e. t-tau, p-tau and Abeta42). However, it is not clear how many cases were actually positive for AD biomarkers based on the reported summary measures of mean and SD in Table 1. Please clarify.

-On the same line, the authors describe that svPPA is associated with TDP-43 type C. This description of the subtype of TDP-43 appears for the first time in the discussion, but was not introduced to the readers earlier. An introduction of the different TDP-43 subtypes would benefit the reader. Moreover, a subset of svPPA cases can present with other pathologies, including AD pathology, which has been described in up to 25 % of svPPA (e.g. see Mesulam et al., Brain 2014 or Grossman Nat Rev Neurol. 2010). Hence, it would be surprising that all n=18 svPPA cases are classified as non-AD. It is necessary to describe the AD biomarker status, and account for this in the analyses of the data, also for the other two PPA variants. Otherwise, one cannot make claims about certain miRNAs being more involved in AD pathophysiology vs.TDP-43 proteinopathy, as currently suggested in the discussion.

-I would suggest that ROC analyses are performed with biomarker proven non-AD versus AD to see how well serum levels of major hits, e.g. miR-133a-3p and miR-106b-5p are able to distinguish between the two groups and/or between clinical groups (e.g. non-AD versus AD; AD-negative svPPA vs. AD-positive lvPPA; or even AD-negative svPPA vs. AD-positive svPPA, the latter depending on sample size).

-Another point is that findings in ALS, a syndrome which is often associated with TDP-43 type A and/or B, not type C, are potentially incorrectly assumed here to translate to findings in svPPA, which is most often a TDP-43 type C proteinopathy. E.g. see De Rossi et al., 2022 for differential effects regarding molecular mechanisms underpinning TDP subtypes. Authors should adapt their discussion accordingly.

-Regarding methodology, DIANA-mirPath has been used for pathway analysis, which included enrichment analyses using the KEGG pathway. The restricted database search might have limited the outcome of the results. E.g. the Database for Annotation, Visualization and Integrated Discovery (DAVID) provides a comprehensive set of several functional annotation tools, including the KEGG database, but also the Reactome pathway, GO terms for molecular and cellular functions, etc. Can the authors perform a more in-depth pathway analysis? What was the reason that no pathway analysis was done for miRNA findings in lvPPA?

-Explain whether ‘mirPath’ also applies any correction for multiple comparisons such as Benjamini-Hochberg. In general, are the reported p-values throughout the Results corrected p-values? The results mention “a statistical cut-off”, but it is not clear what the actual cut-off is for the different analyses applied here based on the Methods section.

-Data in figure 1B and figure 2C and D could potentially be more informative when presented as volcanoplots. Contrasts between each PPA variant and controls are shown, but as the authors claim differential expression between svPPA and lvPPA, a direct contrast between the two could also be informative. This is partly done in figure 3, but I suggest to use a more standardly used volcanoplot for visualization of differential expression.

Minor points

-A measure of sample size (total or per clinical group) could be added to the abstract, as well as a more detailed description of the actual pathway findings.

-The introduction contains a sentence that nfvPPA is a tauopathy: please change to ‘FTLD tauopathy’, as AD is in fact also a tauopathy and not only related to amyloid as posed in the discussion “role in the amyloid metabolism defect”. Adapt these sentences with complete information.

-nfvPPA and svPPA affect the frontal and temporal lobes: here the word ‘respectively’ would be useful as the temporal lobe is initially more affected in svPPA and the frontal lobe relatively more in nfvPPA (e.g. see Gorno-Tempini papers).

-‘No disease-modifying treatments exist’, this is in fact not wrong but note that, in lvPPA with positive AD biomarker status, off-label cholinomimetics are often prescribed (e.g. see Mesulam et al., Neurology, 2019; Schaeverbeke et al., NeuroImage:Clinical, 2017).

-The sentence on miRNAs role in FTD (ref 8-11) contains a wrong reference number 11 without a title.

-The methods could benefit from a clear description on how CSF was analysed: which cut-off values have been used for CSF AD biomarkers? This would be relevant for classifying cases as AD versus non-AD (see main points).

-The labels of the bars in figures 1-2 are difficult to read and often don’t correspond to the miRNAs in the text or caption of figures. Please modify with the correct numbers (e.g. caption figure 1 C: ‘miR-103-3p’ versus figure title ‘miR-133a-3p’).

-The same applies for the results section on nfvPPA and lvPPA, where miRNAs in the text do not appear on the figures (e.g. miR-22-3p). Also, there are errors in the paragraph on pathway analyses: ‘miR-133a-5p’.

-Figure 3 labels are too small and should be plotted in a bigger font.

-Remove the Italian title for the references in Supplementary materials.

Author Response

Point by point replies to criticisms raised by Reviewer #2

Major points

- In this cohort of PPA patients, the methods and Table 1 describe that patients underwent lumbar puncture for AD biomarker analyses (i.e. t-tau, p-tau and Abeta42). However, it is not clear how many cases were actually positive for AD biomarkers based on the reported summary measures of mean and SD in Table 1. Please clarify. As suggested, we have now specified, at the bottom of Table 1, the number of patients having CSF AD signature for each PPA variant, and discussed this issue in the text: “It is important to take in account that a subset of svPPA cases can present with other pathologies, including AD pathology, which has been described in up to 25 % of svPPA [34,35]. Nevertheless, a recent meta-analysis showed that amyloid has been detected (via CSF biomarkers, Amyloid PET or autopsy) in 16% of svPPA [19], which is in line with our results (11%)”.

- On the same line, the authors describe that svPPA is associated with TDP-43 type C. This description of the subtype of TDP-43 appears for the first time in the discussion, but was not introduced to the readers earlier. An introduction of the different TDP-43 subtypes would benefit the reader. Moreover, a subset of svPPA cases can present with other pathologies, including AD pathology, which has been described in up to 25 % of svPPA (e.g. see Mesulam et al., Brain 2014 or Grossman Nat Rev Neurol. 2010). Hence, it would be surprising that all n=18 svPPA cases are classified as non-AD. It is necessary to describe the AD biomarker status, and account for this in the analyses of the data, also for the other two PPA variants. Otherwise, one cannot make claims about certain miRNAs being more involved in AD pathophysiology vs.TDP-43 proteinopathy, as currently suggested in the discussion: more details on TDP43 have been given in the introduction: “Based on histopathology, the accumulation of TDP-43 in the brain cortex is classified into 4 types, type A through D, depending on the forms of TDP-43 accumulation in the cerebral cortex. Type A pathology is frequently observed in FTD and nfvPPA. Type B pathology oc-curs in FTLD with Amyotrophic Lateral Sclerosis (ALS) and ALS with lesions that extend to the cerebral cortex. Type C pathology is associated with svPPA only [4].”. See instead the previous point as regards the presence of CSF AD signature in PPA patients.

- I would suggest that ROC analyses are performed with biomarker proven non-AD versus AD to see how well serum levels of major hits, e.g. miR-133a-3p and miR-106b-5p are able to distinguish between the two groups and/or between clinical groups (e.g. non-AD versus AD; AD-negative svPPA vs. AD-positive lvPPA; or even AD-negative svPPA vs. AD-positive svPPA, the latter depending on sample size). Thank you for the suggestion. We performed ROC curve analyses for miR-133a-3p and miR-106b-5p by comparing fold regulations between groups: controls vs svPPA; svPPA vs other PPA variants; AD-negative svPPA vs AD-positive lvPPA+AD-positive nfvPPA. We included this analysis in the “Results” section: “Receiver operating characteristic curve (ROC) analysis was performed to evaluate the diagnostic potential of miR-133a-3p and miR-106b-5p. In serum, miR-133a-3p together with miR-106b-5p had an AUC of 0,84 to distinguish svPPA from controls (95% CI: 0,7309 to 0,9636, p<0.001, Figure 2A). Moreover, these two miRNAs had an AUC of 0,9696 to dis-tinguish svPPA from other PPA variants (lvPPA+nfvPPA; 95% CI: 0,9338 to 1,000, P<0.001, Figure 2B). Also considering the presence of positive CSF AD biomarkers, serum miR-133a-3p and miR-106b-5p had an AUC of 0,83 to distinguish AD-negative svPPA patients from AD-positive other PPA variants (AD-positive lvPPA+AD-positive nfvPPA; CI: 0,6576 to 1,000, P= 0,0056”, as well as a new Figure (Figure 2C). Methods related to CSF collection and analysis have been included in the “Methods” section.

- Another point is that findings in ALS, a syndrome which is often associated with TDP-43 type A and/or B, not type C, are potentially incorrectly assumed here to translate to findings in svPPA, which is most often a TDP-43 type C proteinopathy. E.g. see De Rossi et al., 2022 for differential effects regarding molecular mechanisms underpinning TDP subtypes. Authors should adapt their discussion accordingly. Thank you for this comment; We adapted the discussion accordingly and included the citation mentioned: “..even if ALS is associate with TDP-43 type A and/or B, not C , that is the TDP-43 subtype of-ten associated with svPPA [4,33].

- Regarding methodology, DIANA-mirPath has been used for pathway analysis, which included enrichment analyses using the KEGG pathway. The restricted database search might have limited the outcome of the results. E.g. the Database for Annotation, Visualization and Integrated Discovery (DAVID) provides a comprehensive set of several functional annotation tools, including the KEGG database, but also the Reactome pathway, GO terms for molecular and cellular functions, etc. Can the authors perform a more in-depth pathway analysis? What was the reason that no pathway analysis was done for miRNA findings in lvPPA? Thank you very much for the suggestion.  We performed a more in depth pathway analysis by using DAVID.  The new results were included in the Results section: “In order to gain understanding of the potential biological significance of these miR-NA deregulations, an in silico approach to identify potential target genes of miR-133a-3p and miR-106b-5p was used. Two publicly available bioinformatic tools were used: Tar-getScan 7.0 and miRDB. As shown in Figure 4A, a total number of 492 overlapping pre-dicted target genes for these miRNAs were identified from these algorithms (Supplemen-tary Table1 and 2). To understand the functional roles and mechanisms of identified tar-get genes, GO and KEEG analyses were performed using DAVID. The results demonstrat-ed that genes were enriched in ten BP, such as, nervous system development and positive regulation of dendrite extension (Figure 4B). While for MF, the genes were enriched in small GTPase binding and in protein binding, for example (Figure 4C). Additionally, CC analysis showed that the genes were enriched, among others, in nucleoplasm, cytoplasm and glutamatergic synapse (Figure 4D). The KEEG pathway analysis predicted that these mRNAs were mainly involved in MAPK signaling pathway, Dopaminergic synapsis and longevity regulating pathway (Figure 4E)”, as well as the “Methods” section, paragraph 2.7: “Target gene prediction and bioinformatics analysis”. Regarding the second question, we chose to perform pathway analysis for the statistically significant miRNAs only, as mentioned in the “Results” section.

- Explain whether ‘mirPath’ also applies any correction for multiple comparisons such as Benjamini-Hochberg. In general, are the reported p-values throughout the Results corrected p-values? The results mention “a statistical cut-off”, but it is not clear what the actual cut-off is for the different analyses applied here based on the Methods section. As suggested by the Reviewer, we performed pathway analysis with DAVID and we deleted the DIANA miR-path analysis since it is less extensive.

- Data in figure 1B and figure 2C and D could potentially be more informative when presented as volcanoplots. Contrasts between each PPA variant and controls are shown, but as the authors claim differential expression between svPPA and lvPPA, a direct contrast between the two could also be informative. This is partly done in figure 3, but I suggest to use a more standardly used volcanoplot for visualization of differential expression. Thank you for the suggestions. We change the type of graph and we included volcanoplots to visualize the differential expression. We also include a volcano plot between svPPA and lvPPA (Figure1A and Figure 3).

Minor points

- A measure of sample size (total or per clinical group) could be added to the abstract, as well as a more detailed description of the actual pathway findings. We included the sample size and the patwhays.

- The introduction contains a sentence that nfvPPA is a tauopathy: please change to ‘FTLD tauopathy’, as AD is in fact also a tauopathy and not only related to amyloid as posed in the discussion “role in the amyloid metabolism defect”. Adapt these sentences with complete information. We include the term “ FTLD tauopathy”.

- nfvPPA and svPPA affect the frontal and temporal lobes: here the word ‘respectively’ would be useful as the temporal lobe is initially more affected in svPPA and the frontal lobe relatively more in nfvPPA (e.g. see Gorno-Tempini papers). We corrected the sentence according to this suggestion.

-‘No disease-modifying treatments exist’, this is in fact not wrong but note that, in lvPPA with positive AD biomarker status, off-label cholinomimetics are often prescribed (e.g. see Mesulam et al., Neurology, 2019; Schaeverbeke et al., NeuroImage:Clinical, 2017). We included this information in the Introduction section: “Excluding off-label cholinomimetics for lvPPA (with positive AD biomarkers), no disease-modifying treatments for PPA variants exist”.

- The sentence on miRNAs role in FTD (ref 8-11) contains a wrong reference number 11 without a title. We corrected the above-mentioned reference.

- The methods could benefit from a clear description on how CSF was analysed: which cut-off values have been used for CSF AD biomarkers? This would be relevant for classifying cases as AD versus non-AD (see main points). We included cut-off values in the Methods section the CSF biomarkers measurement. (Paragraph 2.3).

- The labels of the bars in figures 1-2 are difficult to read and often don’t correspond to the miRNAs in the text or caption of figures. Please modify with the correct numbers (e.g. caption figure 1 C: ‘miR-103-3p’ versus figure title ‘miR-133a-3p’). We changed the type of graphs and figures accordingly with previous comments.

- The same applies for the results section on nfvPPA and lvPPA, where miRNAs in the text do not appear on the figures (e.g. miR-22-3p). Also, there are errors in the paragraph on pathway analyses: ‘miR-133a-5p’.: please see the reply above

- Figure 3 labels are too small and should be plotted in a bigger font. We plotted the figure in bigger font

- Remove the Italian title for the references in Supplementary materials. We are really sorry for the inconvenience; we corrected the title.

Round 2

Reviewer 1 Report

The authors have addressed most of my concerns. I still believe all figures need to be improved extensively (numbering and figures are not aligned, numbers are too small, lack uniformity, font is not always the same, etc.). The table also needs to be slightly improved ("," should be "."). 

Author Response

The authors have addressed most of my concerns. I still believe all figures need to be improved extensively (numbering and figures are not aligned, numbers are too small, lack uniformity, font is not always the same, etc.). The table also needs to be slightly improved ("," should be "."). 

We have tried to improve the quality of the Figures as suggested by the Reviewer.

Reviewer 2 Report

The authors have answered most of my questions/comments. I do suggest some minor corrections:

-The description of TDP-43 subtypes is incomplete: a subtype E has been described since 2017 (Lee et al., DOI 10.1007/s00401-017-1679-9).

-Moreover, TDP type C does not exclusively appear in svPPA: so remove "only" and replace by "mainly".

-The volcanoplots should contain labels for at least a subset of up/downregulated miRNAs, e.g. in Fig1 A, it is not clear where miR-106b-5p and miR-133a-3p are situated. Same for Figure 3.

-The biological processes are only partly shown in Figure 4B as text disappears. Please correct this.

-missing authors for ref 6.

-wrong reference 4: For the FTLD subtypes they should refer to the original paper of Mackenzie et al., 2011 and/or Lee et al., 2017, NOT de Rossi (reference 4 doesn't belong there).

-It is KEGG, not KEEG.

Author Response

The authors have answered most of my questions/comments. I do suggest some minor corrections:

-The description of TDP-43 subtypes is incomplete: a subtype E has been described since 2017 (Lee et al., DOI 10.1007/s00401-017-1679-9).

We have also mentioned the subtype E according to Lee et al., 2017

-Moreover, TDP type C does not exclusively appear in svPPA: so remove "only" and replace by "mainly".

We have corrected as suggested by the Reviewer.

-The volcanoplots should contain labels for at least a subset of up/downregulated miRNAs, e.g. in Fig1 A, it is not clear where miR-106b-5p and miR-133a-3p are situated. Same for Figure 3.  

As suggested, we have tried to improve the quality of the Figures.

-The biological processes are only partly shown in Figure 4B as text disappears. Please correct this.

We have corrected as suggested.

-missing authors for ref 6.

We have checked the Author names.

-wrong reference 4: For the FTLD subtypes they should refer to the original paper of Mackenzie et al., 2011 and/or Lee et al., 2017, NOT de Rossi (reference 4 doesn't belong there).

We have included Mackenzie et al., 2011 and Lee et al., 2017.

-It is KEGG, not KEEG.

We have corrected, as suggested.

Round 3

Reviewer 1 Report

I believe the figures are still poorly formatted but the content is easily understandable.